# Neuropathic Pain in the Emergency Setting: Diagnosis and Management

**DOI:** 10.3390/jcm12186028

**Published:** 2023-09-18

**Authors:** Pietro Emiliano Doneddu, Umberto Pensato, Alessandra Iorfida, Claudia Alberti, Eduardo Nobile-Orazio, Andrea Fabbri, Antonio Voza

**Affiliations:** 1Neuromuscular and Neuroimmunology Unit, IRCCS Humanitas Research Hospital, Via Manzoni 56, 20089 Rozzano, MI, Italy; 2Department of Biomedical Sciences, Humanitas University, Via Rita Levi Montalcini 4, 20072 Pieve Emanuele, MI, Italy; 3Neurology and Stroke Unit, IRCCS Humanitas Research Hospital, Via Manzoni 56, 20089 Rozzano, MI, Italy; 4Emergency Department, IRCCS Humanitas Research Hospital, Via Manzoni 56, 20089 Rozzano, MI, Italy; 5Department of Medical Biotechnology and Translational Medicine, Milan University, 20133 Milano, MI, Italy; 6Emergency Department AUSL Romagna, Presidio Ospedaliero Morgagni-Pierantoni, 47121 Forlì, FC, Italy

**Keywords:** neuropathic pain, pain management, emergency department

## Abstract

Neuropathic pain, traditionally considered a chronic condition, is increasingly encountered in the emergency department (ED), accounting for approximately 20% of patients presenting with pain. Understanding the physiology and key clinical presentations of neuropathic pain is crucial for ED physicians to provide optimal treatment. While diagnosing neuropathic pain can be challenging, emphasis should be placed on obtaining a comprehensive medical history and conducting a thorough clinical examination. Patients often describe neuropathic pain as a burning or shock-like sensation, leading them to seek care in the ED after ineffective relief from common analgesics such as paracetamol and NSAIDs. Collaboration between emergency medicine specialists, neurologists, and pain management experts can contribute to the development of evidence-based guidelines specifically tailored for the emergency department setting. This article provides a concise overview of the common clinical manifestations of neuropathic pain that may prompt patients to seek emergency care.

## 1. Introduction

Pain is a common complaint among patients referred to hospitals, accounting for about 50–80% of emergency department visits [1,2,3]. Traditionally, pain in the emergency setting was believed to be primarily nociceptive, resulting from tissue injury. However, more attention has recently been paid to other types of pain in patients who access hospitals, particularly neuropathic pain. Neuropathic pain is defined as pain caused by a lesion or disease of the somatosensory nervous system [4]. Estimates of its prevalence in the general population vary from 1% to 8% [5,6], while its frequency among patients reporting pain in the emergency department is about 20% [7]. Although commonly referred to as a chronic manifestation, neuropathic pain can initially present as an acute symptom, prompting patients to seek emergency room care and representing a diagnostic challenge for physicians. There are many disorders and conditions that can clinically manifest as neuropathic pain. These include infectious or post-infectious diseases (e.g., HIV/AIDS and postherpetic neuralgia), herniated intervertebral discs compressing nerve roots, chronic medical conditions (e.g., diabetes, cancer, multiple sclerosis, and stroke), trauma (e.g., limb amputation or nerve injuries due to accidents), treatment (e.g., chemotherapy or radiotherapy), or surgery (nerve injuries resulting from medical procedures). These causes can be broadly categorized anatomically into three basic types: (i) peripheral neuropathic pain syndromes, (ii) central neuropathic pain syndromes, and (iii) cranial neuralgia (Table A1).

## 2. Mechanisms of Neuropathic Pain

Pain pathways are among the most important systems that have evolved to ensure the survival of individuals by recognizing pain as a warning sign. Although the exact mechanism behind neuropathic pain is still not fully understood, numerous animal studies have revealed the involved pathways. The receptors responsible for sensing pain are known as nociceptors, which are activated by noxious stimuli such as mechanical, thermal, and chemical factors. These stimuli trigger high-threshold mechanoceptors and polymodal nociceptors, which respond to inputs that can damage tissues. Pain arises from a combination of tissue trauma, inflammation, and direct nerve injury. Nociceptors are found at the ends of pseudounipolar sensory neurons, with their cell bodies located in the dorsal root ganglia, trigeminal ganglion, or nodose ganglia. They terminate in the dorsal horn of the spinal cord. There are two main types of primary afferent fibers that transmit pain signals to the central nervous system:-A-delta fibers: These are small myelinated neurons that transmit rapid and well-localized pain sensations. They have a diameter of 1–5 μm and a conduction velocity of 3–30 m/s.-C fibers: These are smaller, unmyelinated fibers that transmit diffuse and aching pain. They have a diameter of 0.2–1.5 μm and a conduction velocity of 0.5–2.0 m/s.

While nociceptive pain is caused by identifiable lesions that cause tissue damage in somatic or visceral structures, neuropathic pain is an abnormal sensation resulting from direct damage to the central nervous system (CNS) or peripheral nervous system (PNS). The etiologies and mechanisms underlying the onset of neuropathic pain exhibit considerable variability. Table A1 outlines the postulated pathogenetic mechanisms responsible for pain in the disorders associated with neuropathic pain, most frequently encountered in the emergency department.

### 2.1. Peripheral Nerve Injury

Abnormal pain sensations caused by lesions in A-delta or C fibers result from various potential mechanisms. One mechanism involves the fibers’ attempt at regeneration after initial damage or dysfunction, leading to neuroma formation [8]. This regenerative process can alter sodium channel distribution and increase expression, contributing to abnormal sensations such as hyperalgesia (increased sensitivity to painful stimuli) and allodynia (pain in response to non-painful stimuli). Another possible mechanism underlying neuropathic pain is the presence of ectopic impulses due to myelin barrier breakdown from a lesion [9]. The expression of voltage-gated calcium channels can also increase, correlating with the presence of allodynia [10,11]. The high ion channel density can reduce resting potential, causing ectopic impulses. This phenomenon is supported by the analgesic efficacy of calcium-channel antagonists and sodium-channel blockades. After nerve injury, peripheral nociceptors exhibit lower firing thresholds, an increased response to pain stimuli, and can even fire in response to non-noxious stimuli [12].

### 2.2. Central Sensitization

Central sensitization denotes an augmented reactivity to regular or subthreshold inputs, culminating in heightened sensitivity, elevated responsiveness to non-painful stimuli, and an extended receptive field. In instances where sustained noxious stimuli activate C fibers, this process can induce an altered response within the dorsal horn of the spinal cord. This alteration encompasses diminished local inhibition facilitated by GABA and glycine alongside the excitotoxic demise of inhibitory interneurons. Inflowing axons may experience heightened activity within spinothalamic tract neurons due to the involvement of *N*-methyl-d-aspartate (NMDA), nitric oxide (NO), and neurokinins. This cascade subsequently diminishes the pain-signaling threshold [9].

## 3. General Principles of Neuropathic Pain in the Emergency Setting

### 3.1. Evaluation of the Patient with Pain

Defining the distinct characteristics of neuropathic pain in a patient’s medical history is crucial, as the clinical differentiation between neuropathic pain and non-neuropathic pain is essential for a correct diagnosis. Pain descriptors such as “burning”, “pins and needles”, “shooting”, “tingling”, or “electric” can aid in distinguishing neuropathic pain from non-neuropathic pain, which, in contrast, is typically described as “dull”, “throbbing”, “pressure”, or “aching”. In more advanced cases, patients with neuropathic pain may report allodynia (pain from normally non-painful stimuli) or hyperalgesia (an excessive pain response to typically painful stimuli). However, it is important to note that no single descriptor should be considered pathognomonic, as acute pain often presents with a combination of neuropathic and non-neuropathic features. Indeed, approximately 50% of patients with musculoskeletal pain use words commonly associated with neuropathic pain [13]. Although validated screening questionnaires exist for chronic neuropathic pain, their validation for acute-onset presentations is lacking. Additionally, their use in the emergency setting is limited due to patients’ life-threatening clinical conditions requiring immediate medical attention. Consequently, a comprehensive examination is crucial to determining the presence of neuropathic pain amidst ambiguous symptoms and conditions.

Diagnosing neuropathic pain requires demonstrating, during a physical examination, the presence of sensory, autonomic, and/or motor signs occurring in the same distribution as the pain. The examination should also identify the relevant neural pathway responsible for the patient’s symptom distribution (e.g., peripheral nerve lesion, cranial nerve lesion, or spinal lesion).

Patients presenting with peripheral neuropathy typically exhibit more than just pain; they often report localized sensory and motor impairments in specific cutaneous or muscular areas. This may manifest, for instance, as sensations of numbness or tingling in select fingers or toes or in a small region of skin on the face or trunk, all corresponding to the distribution of one or more peripheral nerves. In cases of polyneuropathy, a stocking-glove pain distribution is commonly observed. This distribution can coincide with a reduction in muscular strength in the distal regions of the limbs. In the assessment of a patient with peripheral neuropathy, a physical examination frequently reveals diminished reflexes and flaccid muscle tone.

In contrast, patients with central nervous system disorders typically experience pain alongside sensory and/or motor disturbances that extend over a broader area. For example, such disturbances might encompass both the proximal and distal portions of both lower limbs or affect one side of the body. Additionally, these cases may involve accompanying confusion, altered consciousness, or speech difficulties. In situations involving chronic pathologies, individuals might also display spastic tone and brisk reflexes in the affected limbs, along with sensory and motor disruptions and pain.

The IASP/NeuPSIG (International Association for the Study of Pain/Neuropathic Pain Special Interest Group) definition and grading system requires for the diagnostic category of confirmed neuropathic pain either direct anatomic/surgical evidence of a nerve lesion or some objective confirmatory tests (e.g., ultrasound, MRI, neurophysiological tests, biopsy) [14]. Because of the limited availability of such tests in the emergency department, the diagnosis of neuropathic pain in the acute setting remains mostly clinical. However, instrumental tests may be necessary in selected cases where diagnostic doubts persist.

A pain assessment should include a thorough general medical history and physical examination, a specific history of the pain under evaluation, and an assessment of any associated functional impairment. Special consideration for pain assessment needs to be given to those unable to communicate verbally, including the very young and the elderly—particularly those with cognitive impairment—and patients with a lack of consciousness or language barriers.

The patient’s self-report is the most accurate and reliable evidence of the existence of pain and its intensity, and this holds true for patients of all ages, regardless of communication or cognitive deficits. Simple, self-reported, easy-to-use scales that work within the emergency department clinical environment may be worth considering. These scales enable quickly quantifying the patient’s pain intensity and monitoring the longitudinal course and the response to therapy. Examples of these scales are the NRS (numerical rating scale, range 0–10) and the VAS (visual analog scale, range 0–10) [15,16]. Respondents rate the intensity or severity of each descriptor item on a scale from 0 to 10, with 0 being “no pain” and 10 corresponding to “the most severe sensation imaginable.” The Leeds Assessment of Neuropathic Symptoms and Signs (LANSS) scale was designed to help distinguish neuropathic from nociceptive pain [17]. It has two components: a pain questionnaire and a sensory testing component. The pain questionnaire consists of five items that ask about pain characteristics. The sensory testing component asks a clinician to test for allodynia and for altered pin-prick threshold. Each response is weighted, and the weights of all positive responses are summed to create a total score, with a score of less than 12 indicating no neuropathic pain. Another scale designed to discriminate between neuropathic and non-neuropathic pain is the Neuropathic Pain Diagnostic Questionnaire (DN4) [18]. Patients are asked whether their pain has burning, painfully cold, or electric shock qualities and to indicate if they do (or do not) experience tingling, pins and needles, numbness, or itching in the same area where they experience pain. Finally, the evaluating clinician determines if hypoesthesia to touch or to pin-prick exists in the painful area and whether lightly brushing the area elicits pain. The LANSS and DN4 yielded a high level of accuracy (85–86%) for distinguishing patients with and without neuropathic pain [17,18]. Other scales have been implemented for children or patients with cognitive impairment [19,20]. In the latter patient groups, the observation of signs of pain alongside patient self-reports, including facial expression, heart rate, and respiratory rate, helps confirm the presence of pain.

### 3.2. Treatment of Neuropathic Pain in the Emergency Setting

Neuropathic pain may be the primary reason for admission to the emergency department, especially in patients with high pain intensity or in patients with exacerbations of chronic pain who desire rapid pain control. It may also be one of the components of a broader clinical presentation in a patient entering the emergency department for other reasons (e.g., traumatic limb injury or Guillain–Barré syndrome). In both cases, pain control should not be delayed while waiting for the diagnostic work-up results. The successful management of pain is one of the most important contributions that emergency care practitioners can make because providing pain relief is an important goal, and undertreated pain can have damaging long-term consequences.

The potential sequelae of unmanaged pain for the patient include chronic pain, impacting resources such as increased hospital readmission rates, depression, anxiety, and circulatory system disorders [21]. Studies have shown that patients whose primary pain is well managed and treated in the emergency department have higher overall satisfaction with hospital services [22]. However, there is evidence of inadequate pain treatment in the emergency department, particularly among patients with neuropathic pain, who more frequently have insufficient pain relief at discharge and intense pain [1,2,3]. To our knowledge, no guidelines for the management of neuropathic pain in the emergency department have been published, and therapeutic choice in this clinical setting still relies on drugs that have been tested in clinical trials specifically designed to treat chronic neuropathic pain.

Following the assessment of a patient’s pain, the selection of an appropriate analgesic is crucial, considering both the benefits and risks of pharmacological and non-pharmacological approaches. Non-pharmacological strategies that might be considered in the emergency setting include cryotherapy or heat application to reduce swelling, pain, spasms, and contractures, as well as the proper positioning of fractured limbs. Additionally, providing information and explanations to reassure the patient can be beneficial [13,22]. Medication should be initiated if non-pharmacological measures are insufficient to achieve adequate pain relief. Antidepressants, antiepileptics, topical anaesthetics, and opioids are the most effective pharmacological options. If a patient experiences insufficient relief or adverse effects from a medication, dosage adjustments, alternative medications, or combination therapy can be explored. However, it is important to note that many of these first- and second-line options carry significant potential for side effects, particularly in the elderly.

Topical medications like lidocaine patches and capsaicin may be used as alternatives, although their effectiveness is limited to cases where neuropathic pain is localized to a small area of the skin, such as postherpetic neuralgia or painful length-dependent neuropathies. It is important to recognize that no single drug is universally effective, and pain relief is typically partial. As a result, nearly 50% of patients with neuropathic pain utilize two or more medications, and combination therapy is recognized as an important aspect of neuropathic pain management in most guidelines [23,24,25].

The choice of therapy should also consider factors such as the patient’s pain intensity, the distinction between acute and chronic neuropathic pain, and the pharmacokinetics of the drug. Patients with acute neuropathic pain typically experience high or very high pain intensity, necessitating a pharmacological approach that enables rapid pain control. Conversely, in patients with chronic neuropathic pain, the selection of a drug should consider not only its efficacy but also its long-term sustainability and potential side effects. In many cases, drugs commonly used for chronic neuropathic pain require slow titration of the dosage to avoid side effects that could hinder rapid pain control.

A critical aspect of therapy that is often underestimated is communication with the patient regarding the goals of neuropathic pain therapy. Patients expect a high degree of pain relief, and many expect complete pain resolution. Nonetheless, all the studies suggest that neuropathic pain responds poorly to therapy. In order to promote patient satisfaction, aligning patients’ expectations with the expected efficacy of interventions (approximately 30% pain reduction is considered a success in clinical trials) would be beneficial. This helps prevent the suspension of the drug by the patient, who believes in its ineffectiveness. It is common to encounter patients who have tried various classes of drugs and are no longer on therapy, claiming that none have had much benefit. The goal of neuropathic pain therapy (with some exceptions, for instance, trigeminal neuralgia) is to reduce pain intensity to acceptable levels, allowing the patient to regain a satisfactory quality of life. Often, this can only be achieved through combined therapy and the summation of the effects, albeit slight, of several drugs. Another important recommendation is to prescribe drugs at the maximum tolerated dose before determining their inefficacy.

### 3.3. Prevalence and Causes of Neuropathic Pain in the Emergency Department

No accurate estimate is available regarding the most frequently encountered conditions or diseases causing neuropathic pain in the emergency department. However, a French study conducted at a university hospital’s emergency department revealed that among patients with neuropathic pain, 57% had experienced a traumatic injury, while only 10% had a documented history of chronic pain [7]. Furthermore, 90% of these patients presented with acute pain and had no central neurologic lesions [7]. The study also found that the involvement of limbs was significantly more common [7]. The main pain-causing conditions observed in these patients included open wounds (22%), mechanical low back pain (13%), tendinitis (7%), and arthritis (7%) [7]. These findings are consistent with cross-sectional studies conducted in pain clinics, which identified radiculopathy and neuralgia as the most prevalent causes of neuropathic pain. Another study conducted in Spain reported that 24.5% of patients with neuropathic pain required hospitalization and 44% sought care at the emergency department [7,26], reinforcing the concept that neuropathic pain is not merely a chronic condition.

## 4. Peripheral Neuropathic Pain Syndromes

### 4.1. Postherpetic Neuralgia

The varicella zoster virus, which causes chickenpox, can remain dormant in the sensory ganglion for many years after infection. Its reactivation leads to herpes zoster, a painful rash that typically appears in a single dermatome associated with the affected dorsal root or cranial nerve ganglion. Usually, acute pain precedes the rash by 7–10 days [27]. The rash and pain typically last less than a month [27,28]. However, approximately 10% of patients may develop postherpetic neuralgia as a long-term consequence, wherein neuropathic pain persists in the previously affected area [27,28]. Evidence suggests that the pathogenesis of postherpetic neuralgia involves both peripheral and central mechanisms. Over 95% of young adults in Western countries show seropositivity for antibodies to the varicella zoster virus, putting them at risk of developing this condition [27,28]. Elderly individuals and those with weakened immune systems are more susceptible to developing postherpetic neuralgia. In these patient groups, acute herpes zoster can occur in multiple dermatomes, posing a potentially life-threatening condition [22].

Patients with postherpetic neuralgia typically describe constant deep aching or burning pain with intermittent bouts and allodynia. Some patients also experience itching. The pain is usually localized and one-sided (dermatomal) and is intense enough to disrupt sleep and daily activities. Despite the debilitating nature of this disorder, postherpetic neuralgia is often underdiagnosed and inadequately managed, particularly in primary care settings [27,28], which may result in patients seeking treatment in the emergency department. Although the diagnosis of neuralgia is straightforward in most cases, it can be challenging to recognize in patients where the rash is in an unseen area or those without a visible rash (zoster sine herpete). A physical examination may reveal evidence of skin scarring in the previously affected area, allodynia, or numbness. The presence of antibodies to herpes zoster can support the diagnosis of subclinical herpes zoster infection. Tricyclic antidepressants, gabapentin, pregabalin, and the topical lidocaine 5% patch are considered first-line treatments for postherpetic neuralgia by the American Academy of Neurology (AAN) and the European Federation of Neurological Societies (EFNS/PNS) [23,29]. Opioids, tramadol, capsaicin cream, and patches are typically considered second- or third-line treatments [23,29]. The slow onset of action limits the role of antidepressants and gabapentinoids in acute pain management. The lidocaine 5% patch provides rapid pain relief and excellent tolerability, making it a first-line option in emergency settings and for the elderly [23,29]. The EFNS/PNS guidelines recommend a trial of 2–4 weeks with the lidocaine patch before considering alternative therapies [23]. However, its limited availability in hospitals and emergency departments in some countries restricts its use. Opioids may be considered for acute treatment, although their common side effects restrict their use for chronic pain. For patients who do not respond adequately to single treatments, combination therapy with two agents targeting different action mechanisms may enhance efficacy. This strategy typically involves systemic and topical therapy [16,24]. It is worth mentioning that two vaccines for herpes zoster are licensed and available globally for use in adults aged 50 years or older [30]. These vaccines have demonstrated effectiveness in preventing herpes zoster and reducing the risk of postherpetic neuralgia. They are now recommended for immunocompetent patients aged 60 years and older [30].

### 4.2. Painful Polyneuropathy

Painful polyneuropathy is a broad category encompassing various peripheral neuropathies characterized by the presence of neuropathic pain. The most common form of painful polyneuropathy in the general population is diabetic polyneuropathy, which affects approximately 16% of patients with diabetes [31]. Diabetic polyneuropathy typically has a slow and gradual onset, leading to its diagnosis primarily in outpatient settings. Consequently, patients with diabetic neuropathy rarely seek emergency department care specifically for inadequate pain control or to investigate the cause of their symptoms. Instead, it is more likely that a patient with painful diabetic neuropathy visiting the emergency department for other reasons may report poor pain control. Unfortunately, the pain associated with diabetic polyneuropathy is often undertreated [31]. Studies have shown that female sex and more severe neuropathy are associated with a higher prevalence of pain [31,32,33]. In addition to pain, symptoms of diabetic neuropathy include numbness, tingling, and, less commonly, weakness and unsteadiness. These symptoms typically start distally (at the toes) and progress proximally, eventually involving the upper-limb digits as the lower-limb symptoms extend beyond the knees. Clinical examination findings of diabetic neuropathy may include diminished sensation to pinprick, temperature, vibration, and proprioception in a characteristic “stocking and glove” distribution. The differential diagnosis for painful polyneuropathy is extensive and requires a thorough neurological evaluation.

Another type of polyneuropathy frequently encountered in the emergency department is Guillain–Barré syndrome (GBS), an acute immune-mediated polyneuropathy of post-infectious or post-vaccinal origin. GBS is characterized by progressive motor weakness and areflexia. Although clinicians often prioritize the management of motor weakness due to its impact on daily functioning and potential complications (e.g., respiratory insufficiency), studies have shown that pain is also prevalent in GBS (ranging from 55% to 89% of cases) and can range from moderate to severe in intensity [34]. One study even found that neuropathic pain was associated with a delayed diagnosis of GBS, likely because its frequency is not widely recognized among clinicians and neurologists [35]. However, it is important to note that pain intensity upon admission is not a predictor of poor prognosis in GBS [34,35]. The lower limbs are the most commonly affected site for pain in GBS, followed by the lower back [34,36]. Often, patients may report pain in multiple locations [34,36]. Radicular and muscle pain are the most common types of pain experienced during the acute phase of GBS. The underlying causes of pain in GBS may include inflammation or damage to large myelinated sensory fibers, nerve edema and subsequent compression, and impairment of small nerve fibers [34]. Suspecting GBS is crucial in patients presenting with acute polyneuropathy accompanied by weakness and sensory disturbances. Typical features of GBS include cranial nerve involvement, dysautonomia, and an ascending pattern of symptoms. Absent reflexes are a “red flag” for GBS in patients with rapidly progressive weakness. A recent history of an infectious event is often associated with GBS. A diagnosis ideally requires a neurological evaluation, as studies have shown significantly better outcomes among patients who are assessed by a neurologist during their initial visit [35].

Chemotherapy and certain medications used for cancer treatment can induce peripheral neuropathy, known as “chemotherapy-induced peripheral neuropathy”. Specific chemotherapy drugs, such as platinum, taxanes, vinca alkaloids, and bortezomib (used in myeloma treatment), are more likely to cause neuropathy. Painful chemotherapy-induced peripheral neuropathy typically develops gradually, progressing over weeks or months after the initiation of therapy. It can persist as subacute or chronic pain, which may worsen over time or improve upon discontinuation of therapy. However, in some cases, it may also manifest acutely after the first cycle of chemotherapy, as observed with drugs such as oxaliplatin or paclitaxel [37]. The mechanisms underlying nerve injury in chemotherapy-induced peripheral neuropathy are not fully understood, but the severity of nerve damage appears to be directly related to the drug dosage. The typical clinical presentation of chemotherapy-induced peripheral neuropathy includes ascending distal paraesthesia and dysesthesia, accompanied by burning pain and allodynia in a characteristic “stocking and glove” distribution. The diagnosis of chemotherapy-induced neuropathy relies on recognizing symptoms and a history of symptom onset after initiating chemotherapy. A neurological evaluation is often necessary to rule out other potential causes. Nerve conduction studies are generally not helpful in diagnosing early chemotherapy-induced neuropathy since they primarily reflect axonal loss or demyelination, which typically occur later in the disease progression. These studies do not capture the putative changes in ion channel function or resting membrane potential that contribute to acute chemotherapy-induced neuropathy. Cancer patients may also experience other types of peripheral neuropathies that cause pain, such as direct invasion of peripheral nerves and plexuses by cancer cells or iatrogenic nerve injury resulting from radiotherapy or surgery. In these cases, the patient’s symptoms are typically localized to a specific limb or region of the body that is in direct contact with the tumor mass or has undergone radiotherapy or surgery.

Regarding treatment response, diabetic and non-diabetic peripheral neuropathies can be approached similarly [23]. Tricyclic antidepressants like gabapentin, pregabalin, and selective serotonin–norepinephrine reuptake inhibitors (SNRIs) like duloxetine and venlafaxine are recommended for patients with chronic neuropathic pain [23]. For patients with inadequate control of chronic neuropathic pain, initiating one of these drugs or increasing the dosage of the currently prescribed medication is indicated. In cases of acute exacerbation of chronic pain, acute onset of painful polyneuropathy (e.g., GBS), or coexisting non-neuropathic pain (e.g., cancer invasion of peripheral nerves), short-term use of opioids such as tramadol is recommended due to their rapid onset of action and efficacy in controlling nociceptive pain [23,37,38].

### 4.3. Traumatic Injury of Peripheral Nerves

Traumatic injuries to peripheral nerves occur in approximately 2–5% of patients admitted to trauma centers [39,40,41]. Accidental trauma affecting the peripheral nerves in the upper extremity can have severe consequences, leading to lifelong morbidity, chronic neuropathic pain, and permanent disability, significantly impacting the patient’s quality of life. These injuries predominantly affect young males between 18 and 35 [39,40,41]. In Western countries, most traumatic cases are related to traffic incidents, particularly motorcycle and car collisions [39,40,41]. Blunt trauma is the most common mechanism underlying these injuries, with a smaller proportion resulting from penetrating incidents [39,40,41]. Commonly observed injuries in patients with traumatic nerve injuries include fractures of the humerus or ulna, vascular lacerations, and extensive soft tissue damage [39,40,41]. The radial, ulnar, and median nerves are the most frequently affected in the upper limb, while the sciatic, peroneal, and tibial nerves are commonly involved in the lower limb [39,40,41]. Patients with additional nerve lesions in the context of trauma present with a higher prevalence of shock, longer hospital stays, and a greater need for subsequent inpatient rehabilitation [41]. Nerve injuries can occur as part of a multisystem trauma or as focal injuries. In cases of multisystem trauma, identifying nerve lesions may be more challenging, particularly in blunt trauma.

During the examination of the patient, it is necessary to evaluate the presence of weakness or sensory deficits not only at the site of the trauma but also by extending the examination to the areas unaffected. Traumatic nerve injuries typically manifest with maximal deficits immediately after the injury. A progressive deficit indicates an ongoing secondary process, such as hematoma, edema, or ossifying myositis. Initial management involves addressing acute life-threatening injuries and stabilizing the injuries to prevent further deficits, following standard trauma protocols. Neuropathic pain is prevalent in approximately 50–70% of cases following traumatic nerve injury, and effective pain management is crucial [42,43]. Given that most patients with traumatic nerve injuries also have concomitant injuries to other tissues, they often experience a combination of neuropathic and nociceptive pain. Pain management strategies involve a multimodal approach, combining medications for neuropathic pain, such as tricyclic antidepressants, gabapentin, pregabalin, and SNRIs (duloxetine or nortriptyline), with medications for nociceptive pain, including paracetamol, non-steroidal anti-inflammatory drugs, and opioids [23,44,45].

### 4.4. Back and Neck Pain and Cervical and Lumbosacral Radiculopathies

Back and neck pain are common symptoms experienced by about two-thirds of adults [46]. Low back pain, in particular, is the leading cause of disability and contributes significantly to healthcare costs worldwide [47]. The prevalence of these conditions tends to increase with age. When evaluating back and neck pain, it is important to consider various potential causes, including spine disorders such as compression fractures, spondyloarthropathy, and nerve root compressions, as well as other conditions like malignancy, leaking aortic aneurysms, and visceral diseases. Red flags indicating underlying severe conditions include weight loss, fever, a history of cancer or immunosuppression, recent infection, and concomitant neurological signs and symptoms [46].

In the emergency room, the evaluation of back pain is primarily clinical, and additional tests are generally not required unless a serious condition is suspected. For patients without signs of a severe underlying condition, it is recommended to resume normal activities gradually, continue working, use self-care options (e.g., hot packs) for pain relief, and consider stronger analgesics if necessary [46]. Paracetamol has been found to be ineffective for acute low back pain [48], and the use of opioids for acute pain has not been extensively studied [41], although real-life data indicate their frequent use [49].

The 2016 UK National Institute for Health and Care Excellence (NICE) guidelines suggest two options for managing low back pain: oral NSAIDs at the lowest effective dose for the shortest duration possible and the use of weak opioids (with or without paracetamol) if the patient does not respond well to or cannot tolerate NSAIDs [50]. Upon discharge, non-pharmacological therapies such as manual therapy, exercise, massage, and acupuncture should be recommended [46]. For persistent back pain, systematic reviews support the effectiveness of NSAIDs and opioids, but not paracetamol, muscle relaxants, tricyclic antidepressants, or neuromodulators like gabapentin or pregabalin [46,51,52]. Axial pain often coexists with radicular pain, or radiculopathy, which occurs when a nerve root is compressed.

Patients with radiculopathy may experience radiating pain, sensory disturbances such as hypoesthesia or paraesthesia, reduced osteotendinous reflexes, and, less commonly, weakness in the affected root distribution area. Radicular compression can be caused by conditions such as herniated discs, spondylosis, trauma, instability, or, rarely, tumors. The diagnosis of radiculopathy is primarily clinical, relying on the patient’s history and physical examination. Imaging tests such as MRI and CT scans, as well as electrophysiologic tests, may be used to confirm the diagnosis and rule out other potential causes, including mononeuropathies, polyneuropathies, spinal canal stenosis, and osteoarticular conditions like hip osteoarthritis. These tests are particularly useful when conservative measures prove ineffective, surgery is being considered, or severe progressive neurological signs and symptoms are present. It is worth noting that reported MRI findings for spine disorders can vary significantly [53], leading many physicians to independently verify the reported findings to determine the most appropriate treatment plan. The estimated prevalence of radiculopathy is approximately 10 cases per 1000 individuals [54,55].

Patients with pain related to spinal disorders may experience two types of pain: nociceptive pain resulting from mechanical deformation and inflammation of the bone and contracture of the paraspinal muscles, and neuropathic pain associated with radicular compression. The frequency of neuropathic pain in spinal disorders is high, with some studies reporting rates of around 50%, which surpasses the figures found in conditions like diabetic neuropathy, postherpetic neuralgia, and stroke [56,57]. However, it is important to note that only about 60% of patients with nerve root injuries experience neuropathic pain, suggesting the involvement of mechanisms other than compression in pain generation [57]. The mechanism of nerve root pain is believed to be related to vascular congestion, inflammation, and peri- and intraradicular fibrosis, although pressure may also contribute. There is inconsistency among guidelines regarding pharmaceutical interventions for pain in radiculopathies [58]. Some guidelines suggest NSAIDs, paracetamol, and opioids as options, while others do not [58]. The EFNS/PNS guidelines recommend pregabalin as a first-line treatment and transcutaneous electrical nerve stimulation (TENS) and opioids or their combinations as a second-line option [23].

## 5. Central Neuropathic Pain Syndromes

Even though neuropathic pain usually arises from a dysfunctional peripheral nervous system, it rarely develops following a central nervous system impairment, namely central Neuropathic Pain (CNP) [59,60]. CNP can result from any injury within the central somatosensory nervous system, yet the most common etiologies are stroke, multiple sclerosis, and spinal cord injury.

The diagnosis of CNP can be challenging. Indeed, comorbid peripheral neuropathic pain and nociceptive pain are frequent in patients affected by the abovementioned disorders. Nociceptive pain includes musculoskeletal pain, spasticity-related pain, and overuse pain syndrome in the non-affected limbs. Therefore, meticulous medical history collection and examination are pivotal. The definition of CNP requires two conditions: (i) a spinothalamic tract or thalami lesion anatomically congruent with pain distribution, and (ii) a spinothalamic dysfunction clinically demonstrated with altered pin-prick and/or temperature sensations [59,60]. Only a subgroup of patients with spinal–thalamic–cortical pathway impairments will develop CNP [61]. Indeed, the anatomical lesion is a necessary but insufficient factor for CNP. Surviving spinothalamic axons must develop neuronal hyperexcitability resulting from either denervation hypersensitivity or a lack of descending disinhibition signals [62,63]. Spontaneous burst activity in the deafferented ventrolateral thalamic neurons has been documented with microelectrode recording, suggesting dysfunctional survived spinothalamic circuits as the biological underpinning of central pain generation [64]. Accordingly, patients who experienced complete spinal cord injuries did not develop CNP [65].

Similar to peripheral neuropathic pain, CNP usually presents as a continuous and/or paroxysmal electrical/burning pain of severe intensity. Nonetheless, CNP usually occurs months or even years after the original brain or spinal cord injury, may present as an itching sensation, and is more treatment-refractory than peripheral neuropathic pain [59].

### 5.1. Central Post-Stroke Pain

Stroke is the most common cause of CNP, considering its high incidence worldwide. The incidence and time of onset of CNP following stroke are highly heterogeneous [60]. Indeed, the development of CNP has been observed as far back as three years after the index stroke event, yet it most commonly occurs after a few weeks or months. One study reported an overall incidence of chronic pain in 40% of stroke survivors after four years, yet only 7.3% exhibited CNP [66]. Nonetheless, strategic stroke lesions can significantly increase that percentage. Indeed, CNP is observed in 18% of patients with somatosensory deficits [67] and in up to 25% of patients with ventral posterior nuclei thalamic lesions (Dejerine–Roussy or thalamic pain syndrome), where the pain is distributed in the contralateral body and/or ipsilateral face [68]. CNP is also frequently observed following lateral medullary infarction (pseudo-thalamic pain syndrome) [60]. Interestingly, right-sided thalamic stroke is more associated with CNP, potentially reflecting the right hemisphere’s contribution to pain perception [68].

### 5.2. Multiple Sclerosis-Related Neuropathic Pain

Half of MS patients are burdened by pain symptoms, with 12–28% experiencing CNP [59,69]. Pain can have a patchy distribution, reflecting the multifocality of demyelinating lesions. Risk factors for developing CNP include a progressive phenotype (rather than relapse-remitting), older age, longer disease duration, and higher disease burden [70]. Notably, MS patients have distinctive pain syndromes that can mimic CNP. The Lhermitte sign is a paroxysmal descending electric-like sensation triggered by neck flexion associated with a cervical spine demyelinating plaque in the posterior column [71]. Another common manifestation of MS is painful tonic spasms that are paroxysmal posturing resulting from pyramidal tract lesions [72]. Finally, trigeminal nuclei or trigeminal nerve root entry zone demyelinating plaques are responsible for a prevalence that is 20 times higher compared to the general population of trigeminal neuralgia [73]. These painful neurological manifestations of MS arguably share biological mechanisms. Nonetheless, trigeminal neuralgia and Lhermitte sign have an underlying lesion within the somatosensory pathways (not the spinothalamic, though they may be qualified as atypical central neuropathic pains). Conversely, painful tonic spasms do not involve impairment of the spinal–thalamic–cortical pathways; therefore, they do not qualify as CNP [59].

### 5.3. Spinal Cord Injury-Related Central Neuropathic Pain

Spinal Cord Injury (SCI) may present with pain distributed below or at the level of the anatomical lesions, defined as “below-level” and “at-level” pain [74]. While both pain types are neuropathic, the former is undoubtedly a CNP, whereas the latter can be either central (dorsal horn lesion) or peripheral (root lesion). The differential diagnosis between central and peripheral neuropathic pain in “at-level” pain is usually not straightforward unless an apparent injury of the peripheral nerve/root is evident. CNP occurs in up to 50% of patients following spinal cord lesions and is one of the most burdensome symptoms related to myelopathies [75].

“Below-level” and “at-level” pain following SCI have similar prevalence [76], with the former usually presenting later and more refractory, corroborating its central neuropathic nature. Spinal cord injury has the longest average interval between insult and pain presentation among the CNP syndromes, with symptoms that may appear up to five years after the injury, yet usually within one year [77].

### 5.4. General Treatment Principles in Central Neuropathic Pain

CNP is highly refractory to treatment, often requiring a multimodal pharmacological approach or surgical therapy. Therefore, improving 30–50% of the pain burden is usually a more realistic goal. The small number of randomized clinical trials in CNP results in extrapolating data from studies conducted in other neuropathic pain syndromes. However, treatment responsiveness and efficacy are likely different [78]. In central post-stroke neuropathy, amitriptyline [79] and lamotrigine [80] have proven efficacy, whereas conflicting results have emerged from pregabalin [59]. In MS-related central pain, duloxetine, levetiracetam [81], and cannabinoids have been found to be effective [82,83]. Following spinal cord injury, trials have shown benefits from lamotrigine, botulinum toxin, and pregabalin [84,85,86], whereas conflicting results have been observed for amitriptyline and gabapentin [59].

Surgical lesioning of the spinal–thalamic–cortical pathways has also been tested in CNP. Yet, considering the transitory benefit demonstrated and the highly surgical-related risks, they are not routinely recommended to patients. Conversely, neuromodulation has recently shown promising results in pain control and may have a prominent role in future CNP management.

## 6. Cranial Neuralgia

Cranial neuralgia is an umbrella term that encompasses several disorders characterized by paroxysmal attacks of neuropathic pain localized within a cranial nerve distribution. They are classified according to the International Classification of Headache Disorders Third Edition (ICHD-3) [87]. The most common cranial neuralgias are trigeminal, glossopharyngeal, nervus intermedius, and occipital neuralgia. The term neuralgia refers to the neuropathic quality of pain and differs from the term neuropathy, which defines sensory and/or motor deficits associated with a cranial nerve lesion [88]. These disorders are usually mediated by peripheral neuropathic pain, but lesions within the central cranial nerve pathways rarely underlie the clinical manifestations. Therefore, the neuropathic pain is central [87]. Clinical diagnostic criteria are shared among different cranial neuralgias and require the presence of all the following features: (i) recurrent paroxysmal unilateral pain attacks lasting from a few seconds to two minutes; (ii) severe intensity; (iii) neuropathic quality of pain; and (iv) precipitation by sensory-tactile stimulation within the culprit’s nerve [87]. The anatomical distribution of the pain attacks and the sensory triggers distinguish the different cranial neuralgias. The ICHD-3 further classifies cranial neuralgias based on their underlying aetiologies [87]. Since the most common cause is neurovascular compression, cranial neuralgia is defined as classical when such an abnormality is documented. Conversely, the terms idiopathic and secondary are reserved for conditions when the underlying cause is either not found or different from neurovascular compression, respectively [87,89]. The most common secondary causes of cranial neuralgia include trauma, multiple sclerosis, herpes zoster infection, cerebellopontine angle tumors, and Arnold–Chiari malformations [87].

### 6.1. Trigeminal Neuralgia (TN)

TN is the most common cranial neuralgia, yet it remains a rare disorder with an estimated prevalence of 0.07% [90]. The classical type accounts for 75% of TN and is diagnosed whenever a trigeminal neurovascular compression determining morphological changes to the ipsilateral side is observed either during surgery or on MRI-specific sequences [91]. By definition, TN attacks can be elicited by innocuous mechanical stimulation of the oral mucosa or face innervated by the trigeminal nerve [92]. The most common triggers include light touch, chewing, talking, shaving, washing, or brushing teeth [89]. Both spontaneous and triggered pain attacks are usually followed by a so-called refractory period, during which a further attack cannot be provoked [93]. The pain can be distributed in either one, two, or all three divisions of the trigeminal nerve, yet the maxillary (V2) and mandibular (V3) branches are the most affected, while the ophthalmic nerve (V1) is rarely involved and usually not alone [94]. Associated cranial autonomic ipsilateral manifestations, such as lacrimation, ptosis, miosis, and conjunctival injection, can be observed in one-third of TN patients, usually when the ophthalmic nerve is involved. However, they are usually mild compared to trigeminal autonomic cephalalgias [94]. Up to 50% of patients develop continuous neuropathic pain [89]. It is pivotal to recognize this clinical manifestation since it has different neurobiological underpinnings and responsiveness to treatments compared to paroxysmal neuropathic pain attacks [89]. Differential TN diagnosis is broad and often challenging, including dental causes and trigeminal autonomic cephalalgias. Indeed, a significantly delayed diagnosis and appropriate treatment are often experienced by TN patients. The cutaneous triggerability and refractory periods, along with the pain distribution, are essential features that help diagnose TN correctly [88,89].

In classical TN, neurovascular compression affects the trigeminal root entry zone, which is the most vulnerable portion of the trigeminal nerve and is located near the brainstem [91,95]. This compression results in morphological changes, namely focal demyelination and remyelination, that lower the excitability threshold and promote inappropriate ephaptic propagation from tactile sensory fibers (innocuous mechanical stimuli) to nociceptive fibers (provoked paroxysmal neuropathic pain attacks) [91,96]. Over time, these repetitive stimulations may result in the sensitization of second-order neurons of the trigeminal nuclei that are thought to be responsible for the concomitant continuous neuropathic pain [97]. A brain MRI that includes sequences to examine the cisterns and an MR angiography is essential in the diagnostic work-up of trigeminal neuralgia for excluding secondary causes and detecting potential neurovascular compression [88,89].

The mainstay of TN management is preventive pharmacological therapy, whereas surgical interventions are reserved for patients who do not tolerate or respond to medical treatment.

Treatment guidelines were provided by the European Academy of Neurology in 2019 [98]. However, high-quality studies are lacking, and most recommendations are based on clinical expertise. Carbamazepine or oxcarbazepine are considered first-line therapies and are effective in about 90% of patients in the short term, yet the response rate reduces to 50% in the long term [98,99]. Additionally, these drugs are burdened by several adverse events that lead to withdrawal due to low tolerance in up to 40% of patients [100]. Other drugs that have proven to be effective and are therefore considered second-line options in TN include anti-seizure medications (phenytoin, lamotrigine, and lacosamide), gabapentinoids (pregabalin and gabapentin), botulinum toxin, and lidocaine [98]. Nonetheless, some authors advocate that in cases of refractoriness to carbamazepine or oxcabarzamzepine at adequate dosages, no other drugs will be effective, and surgery should be pursued [88]. Concomitant continuous neuropathic pain has different pathological mechanisms and does not usually respond to classical anti-seizure medications, while gabapentinoids are usually more effective [89].

Surgical interventions are considered disease-modifying treatments as they act directly on the underlying mechanisms and can be curative. Three different surgical options are available for TN patients: (i) stereotactic surgery; (ii) invasive–ablative therapies (balloon compression and radiofrequency thermocoagulation); and (iii) microvascular decompression. The latter is the first-choice surgery in patients with classical TN and is curative in up to 90% of patients [89,98]. Notably, emerging evidence supports its efficacy in non-classical TN [89,98].

A severe acute exacerbation of TN represents a unique clinical scenario that should be discussed separately. In this condition, the pain is persistent and burdens remarkable daily activities such as eating and talking. In this context, hospital admission to provide prompt treatment is usually required [98]. Oral drug titration might result in a remarkable delay in pain relief. Therefore, intravenous administration of drugs such as fosphenytoin, lacosamide, and lidocaine is necessary [98,101,102,103]. Considering the severity of symptoms and the need for intravenous administration of anti-seizure medications at high doses that characterized this clinical presentation, similar to status epilepticus, we believe the definition of “trigeminal neuralgia status” is appropriate.

### 6.2. Glossopharyngeal Neuralgia

Glossopharyngeal neuralgia is a cranial neuralgia characterized by paroxysmal attacks of neuropathic pain in the glossopharyngeal nerve territory (jaw angle, ear, pharynx, and posterior tongue) [104]. It is an extremely rare condition, with an estimated incidence of 1 case per 200,000 individuals [105]. Remarkably, glossopharyngeal neuropathic pain attacks may be accompanied in 2% of patients by syncope mediated by excessive vagal nerve involvement, a condition termed vasoglossopharyngeal neuralgia [106]. Secondary causes of glossopharyngeal neuralgia include neurovascular compression and demyelinating plaques, similar to TN, but also parapharyngeal abscesses and elongated styloid processes (Eagle syndrome) [104]. Management is similar to TN and comprises both pharmacological and surgical treatments.

### 6.3. Nervus Intermedius Neuralgia

Nervus intermedius neuralgia is an extremely rare cranial neuralgia characterized by paroxysmal neuropathic pain attacks in the deep ear elicited by auricular/periauricular mechanical stimuli, sometimes accompanied by ipsilateral autonomic manifestations (taste, salivation, or lacrimation) [87]. Since the nervus intermedius is a branch of the facial nerve, secondary causes of nervus intermedius neuralgia include facial nerve inflammation disorders such as Ramsay–Hunt syndrome or Bell’s palsy [87,107].

### 6.4. Occipital Neuralgia

Occipital neuralgia is an extremely rare cranial neuralgia presenting with paroxysmal neuropathic pain attacks in the posterior skull base that may radiate up to the vertex [87]. Distinctive secondary causes include Arnold–Chiari malformations and upper cervical cord lesions [108]. Management is similar to other cranial neuralgias, yet nerve blocks with a combination of anaesthetics and steroids might also be performed [108].

## 7. Novel Therapeutic Approaches and Targets

Several cutting-edge preclinical and clinical studies have been conducted to explore the efficacy of novel drugs that are expected to expand our therapeutic armamentarium in neuropathic pain. These include, but are not limited to, anti-*N*-methyl-d-aspartate receptor (NMDAR) antagonists, cannabis sativa derivates, non-invasive transcranial brain stimulation, and spinal cord stimulation [109,110,111,112,113]. Additionally, new compounds and therapeutic targets for neuropathic pain are being explored in preclinical studies, including natural compounds, receptors expressed in the microglia, supraspinal gabaergic systems, and ion channels expressed in the dorsal and trigeminal root ganglia, such as Transient Receptor Potential Ankyrin 1 (TRPA1) and sodium channels [110,111,112,113,114,115,116,117,118,119].

## 8. Conclusions

Neuropathic pain presents a significant challenge in the emergency department due to its complex etiology and the limitations of current diagnostic and treatment approaches. It is crucial for healthcare professionals in the emergency setting to have a comprehensive understanding of neuropathic pain to ensure appropriate management. Implementing standardized assessment tools and incorporating a detailed patient history, along with a thorough physical examination, can aid in identifying neuropathic pain etiologies. Pharmacological interventions remain the mainstay of treatment for neuropathic pain in the emergency department. However, integrating a multidisciplinary approach that combines various modalities may optimize pain relief and improve patients’ overall well-being. Further research is needed to advance our understanding of neuropathic pain and explore novel therapeutic strategies.

## Data Availability

No new data were created or analyzed in this study. Data sharing is not applicable to this article.

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
