# Peer review of "Neuropathic Pain in the Emergency Setting: Diagnosis and Management"

_jcm, 2023, doi:10.3390/jcm12186028_

Round 1
Reviewer 1 Report
Dear Authors,
I have read your paper on neuropathic pain with interest. As it is not that common in emergency department specialists in Emergency Medicine are not that aware of this kind of pain. That is why your paper is very useful, nevertheless I have suggestions to improve the educational impact of the paper.
1. Please, add a Table to Your paper with most common causes of acute and chronic of neuropathic pain.
2. Please, provide the reader with information, what elements of neurological assessment should be added to examination of patients with neuropathic pain (e.g. muscle weakness, sensory deficits) with clues upon further diagnostic work-up (e.g. - neuropathic pain + flaccid paresis + acute begining - think of acute inflammatory poluneuropathy; "old" spastic paresis + neuropathic pain - think of central neuropathic pain after stroke, etc.)
3. Please, provide a Table with potential neurological "red flags" for emergency Physician to start rapid diagnostic work-up in Emergency Department (e.g. data from history of physical examination suggestive of malignancy, indication for ugent neuroimaging or lumbar puncture).
Author Response
We thank reviewer 1 for her/his valuable comments
- I have read your paper on neuropathic pain with interest. As it is not that common in emergency department specialists in Emergency Medicine are not that aware of this kind of pain. That is why your paper is very useful, nevertheless I have suggestions to improve the educational impact of the paper. Please, add a Table to Your paper with most common causes of acute and chronic of neuropathic pain. We thank reviewer 1 for her/his suggestion. We have now included a Table where we have listed the most common causes (and their relative mechanisms) of neuropathic pain encountered in emergency department. As explained in the legend, we preferred not to classify the causes in “acute” and “chronic” as most of these disorders may present both with acute and chronic pain
- Please, provide the reader with information, what elements of neurological assessment should be added to examination of patients with neuropathic pain (e.g. muscle weakness, sensory deficits) with clues upon further diagnostic work-up (e.g. - neuropathic pain + flaccid paresis + acute begining - think of acute inflammatory poluneuropathy; "old" spastic paresis + neuropathic pain - think of central neuropathic pain after stroke, etc.)· We agree with reviewer 1. We have now included the following paragraph: “Patients presenting with peripheral neuropathy typically exhibit more than just pain; they often report localized sensory and motor impairments in specific cutaneous or muscular areas. This may manifest, for instance, as sensations of numbness or tingling in select fingers or toes, or in a small region of skin on the face or trunk, all corresponding to the distribution of one or more peripheral nerves. In cases of polyneuropathy, a stock-ing-glove pain distribution is commonly observed. This distribution can coincide with a reduction in muscular strength in the distal regions of the limbs. In the assessment of a patient with peripheral neuropathy, a physical examination frequently reveals dimin-ished reflexes and flaccid muscle tone. In contrast, patients with central nervous system disorders typically experience pain alongside sensory and/or motor disturbances that extend over a broader area. For ex-ample, such disturbances might encompass both the proximal and distal portions of both lower limbs, or affect one side of the body. Additionally, these cases may involve ac-companying confusion, altered consciousness, or speech difficulties. In situations in-volving chronic pathologies, individuals might also display spastic tone and brisk reflexes in the affected limbs, alongside the sensory and motor disruptions and pain.”
- Please, provide a Table with potential neurological "red flags" for emergency Physician to start rapid diagnostic work-up in Emergency Department (e.g. data from history of physical examination suggestive of malignancy, indication for ugent neuroimaging or lumbar puncture).· We thank reviewer 1 for her/his suggestion. We agree with the reviewer that it would be useful for the reader to know the red flags that can help the diagnosis or indicate the need for a rapid work up in conditions that put the patient's life at risk. However, of the disorders listed in the article, only GBS syndrome requires a rapid work up. The other diseases (post herpetic neuralgia, diabetic or small fiber neuropathy, post-stroke painful syndrome) causing neuropathic pain are mostly chronic pathologies that can lead sometimes to an acute exacerbation of pain. Therefore, we believe that this type of table might be not entirely appropriate. We have now included the following sentence in the GBS paragraph: “Absent reflexes are a “red flag” for GBS in patients with rapidly progressive weakness.”
Reviewer 2 Report
The article needs major revision. In this manuscript, the following questions are needed to be addressed:
1. The title and theme of this article are not new and lack of innovation. Several related articles have been published, such as “Neuropathic pain: diagnosis, pathophysiological mechanisms, and treatment”, “Neuropathic Pain: From Mechanisms to Treatment”, suggest an overhaul of the article, and explain how your article differs from the published article, what is the necessity of publishing similar articles? The whole article can be narrowed down to a specific disease to write as a topic. This disease has not been published by others, or the main idea of the article has not been published.
2. The article from the following aspects, Mechanism of neuropathic pain, General principles of neuropathic pain in the emergency setting, Peripheral Neuropathic Pain syndromes, Central Neuropathic Pain Syndromes, Cranial Neuralgia, summarize the related knowledge of neuropathic pain. This article introduces various types of neuropathic pain, and the therapeutic drugs recommended by the guidelines of international authoritative organizations. The description is comprehensive, but the whole article is more of a simple narrative knowledge, without a profound discussion of neuropathic pain related knowledge.
3. Please add the mechanism figure of neuropathic pain.
4. Please make a summary figure based on the content of the whole article.
5. Please add a comprehensive Pain assessment scale, such as LANSS、DN4、ID Pain.
6. The lack of discussion on the treatment of neuropathic pain, for example, maybe increase the discussion of ongoing clinical trials.
7. Please refer to more literatures in recent 3-5 years with high impact factors as references.
Moderate editing of English language required
Author Response
We thank reviewer 2 for her/his valuable comments.
1. The title and theme of this article are not new and lack of innovation. Several related articles have been published, such as “Neuropathic pain: diagnosis, pathophysiological mechanisms, and treatment”, “Neuropathic Pain: From Mechanisms to Treatment”, suggest an overhaul of the article, and explain how your article differs from the published article, what is the necessity of publishing similar articles? The whole article can be narrowed down to a specific disease to write as a topic. This disease has not been published by others, or the main idea of the article has not been published. The article from the following aspects, Mechanism of neuropathic pain, General principles of neuropathic pain in the emergency setting, Peripheral Neuropathic Pain syndromes, Central Neuropathic Pain Syndromes, Cranial Neuralgia, summarize the related knowledge of neuropathic pain. This article introduces various types of neuropathic pain, and the therapeutic drugs recommended by the guidelines of international authoritative organizations. The description is comprehensive, but the whole article is more of a simple narrative knowledge, without a profound discussion of neuropathic pain related knowledge.
- We thank reviewer 2 for her/his comment and suggestion. This is an invited review for a special issue and the title was chosen by the journal. We agree that there are in literature other papers on this topic. However, these articles mostly deal with chronic neuropathic pain or are directed at neurologists. The content of the article carries a practical orientation as it is directed toward physicians and medical personnel undergoing training or working within the field of emergency medicine
2. Please add the mechanism figure of neuropathic pain.
We thank reviewer 2 for her/his valuable suggestion. We believe that given the diversity of causes and mechanisms involved in the genesis of pain in the various disorders mentioned in the article, a table would be clearer and more exhaustive than a figure. We have now included a Table (see reviewer 1) where we have listed the most common causes (and their relative mechanisms) of neuropathic pain encountered in the emergency clinical setting.
3. Please make a summary figure based on the content of the whole article.· We thank reviewer for her/his suggestion. We believe that a summary figure is not strictly necessary for an article of this type, relatively simple and narrative.
4. Please add a comprehensive Pain assessment scale, such as LANSS、DN4、ID Pain.· We agree with reviewer 2 that a paragraph on pain assessment scale would be of interest. We have now included the following paragraph: “Examples of these scales are the NRS (numerical rating scale, range 0-10) and the VAS (visual analogue scale, range 0-10) [15,16]. Respondents rate the intensity or severity of each descriptor item on a scale from 0 to 10, with 0 being “no pain” and 10 corresponding to “the most severe sensation imaginable.” The Leeds Assessment of Neuropathic Symptoms and Signs scale (LANSS) was designed to help distinguish neuropathic from nociceptive pain [17]. It has two components: a pain questionnaire and a sensory testing component. The pain questionnaire consists of five items that ask about pain characteristics. The sensory testing component asks a clinician to test for allodynia and to test to altered pin-prick threshold. Each response is weighted, and the weights of all positive responses are summed to create a total score, with a score of less than 12 indicating an unlike neuropathic pain. Another scale designed to discriminate between neuropathic and nonneuropathic pain is the Neuropathic Pain Diagnostic Questionnaire (DN4) [18]. Patients are asked whether their pain has burning, painfully cold, or electric shock qualities and to indicate if they do (or do not) experience tingling, pins and needles, numbness, or itching in the same area that they experience pain. Finally, the evaluating clinician determines if hypoesthesia to touch or to pin-prick exists in the painful area, and whether lightly brushing the area elicits pain. The LANSS and DN4 yielded a high level of accuracy (85-86%) for distinguishing patients with and without neuropathic pain [17,18]. Other scales have been implemented for children or patients with cognitive impairment [19,20]. In the latter patient groups, also the observation of signs of pain alongside patient self-reports, including facial expression, heart rate, and respiratory rate, help confirm the presence of pain.”
5. The lack of discussion on the treatment of neuropathic pain, for example, maybe increase the discussion of ongoing clinical trials.· We thank Reviewer 2 for this comment. We added a brief subchapter discussing the most cutting-edge research in neuropathic pain management that is expected to change the clinical practice in the next future. Novel Therapeutic Approaches and Targets Several cutting-edge preclinical and clinical studies have been conducted exploring the efficacy of novel drugs that are expected to expand our therapeutic armamentarium in neuropathic pain. These include but are not limited to, anti-N-methyl-D-aspartate re-ceptor (NMDAR) antagonists, cannabis sativa derivates, noninvasive transcranial brain stimulation, and spinal cord stimulation [109-113]. Additionally, new compounds and therapeutic targets of neuropathic pain are being explored in preclinical studies, in-cluding natural compounds, receptors expressed in the microglia, supraspinal gabaergic systems, and ion channels expressed in the dorsal and trigeminal root ganglia such as the Transient Receptor Potential Ankyrin 1 (TRPA1) and sodium channels [110-114].
6. Please refer to more literatures in recent 3-5 years with high impact factors as references.· As requested by Reviewer 2, we added some recent relevant references covering the topic of neuropathic pain (see Ref n 109-119) In case Reviewer 2 believes that some pivotal references are still missing, we are willing to evaluate any specific suggestion.
Round 2
Reviewer 2 Report
Most of the questions are addressed.
minor editing required.